# Four Weeks of Aerobic Training Affects Cardiac Tissue Matrix Metalloproteinase, Lactate Dehydrogenase and Malate Dehydrogenase Enzymes Activities, and Hepatorenal Biomarkers in Experimental Hyperhomocysteinemia in Rats

**DOI:** 10.3390/ijms22136792

**Published:** 2021-06-24

**Authors:** Dusan Todorovic, Marija Stojanovic, Ana Medic, Kristina Gopcevic, Slavica Mutavdzin, Sanja Stankovic, Dragan Djuric

**Affiliations:** 1Faculty of Medicine, Institute of Medical Physiology “Richard Burian”, University of Belgrade, 11000 Belgrade, Serbia; t.dusan@hotmail.com (D.T.); mrj.stojanovic@gmail.com (M.S.); slavica.mutavdzin@gmail.com (S.M.); 2Faculty of Medicine, Institute of Chemistry in Medicine “Prof. Dr. Petar Matavulj”, University of Belgrade, 11000 Belgrade, Serbia; medicana89@gmail.com (A.M.); kristinagopcevic@yahoo.com (K.G.); 3Centre of Medical Biochemistry, Clinical Centre of Serbia, 11000 Belgrade, Serbia; sanjast2013@gmail.com

**Keywords:** homocysteine, exercise, lactate dehydrogenase, malate dehydrogenase, matrix metalloproteinase, hepato-renal markers, heart, rat

## Abstract

The aim of this study was to investigate the effect of the application of homocysteine as well as its effect under the condition of aerobic physical activity on the activities of matrix metalloproteinases (MMP), lactate dehydrogenase (LDH) and malate dehydrogenase (MDH) in cardiac tissue and on hepato-renal biochemical parameters in sera of rats. Male *Wistar albino* rats were divided into four groups (*n* = 10, per group): C: 0.9% NaCl 0.2 mL/day subcutaneous injection (s.c.); H: homocysteine 0.45 µmol/g b.w./day s.c.; CPA saline (0.9% NaCl 0.2 mL/day s.c.) and a program of physical activity on a treadmill; and HPA homocysteine (0.45 µmol/g b.w./day s.c.) and a program of physical activity on a treadmill. Subcutaneous injection of substances was applied 2 times a day at intervals of 8 h during the first two weeks of experimental protocol. Hcy level in serum was significantly higher in the HPA group compared to the CPA group (*p* < 0.05). Levels of glucose, proteins, albumin, and hepatorenal biomarkers were higher in active groups compared with the sedentary group. It was demonstrated that the increased activities of LDH (mainly caused by higher activity of isoform LDH2) and mMDH were found under the condition of homocysteine-treated rats plus aerobic physical activity. Independent application of homocysteine did not lead to these changes. Physical activity leads to activation of MMP-2 isoform and to increased activity of MMP-9 isoform in both homocysteine-treated and control rats.

## 1. Introduction

Cardiovascular diseases (CVD) are responsible for around one third of all deaths worldwide, but this prevalence is still rising [1]. According to the traditional understanding of the pathophysiology of CVD, the main risk factors for its development are smoking, hypertension, sedentary lifestyle and hyperlipidemia, however, data from the literature recognize other risk factors or markers, such as elevated serum homocysteine (Hcy), which should be taken into consideration when assessing the risk of development of CVD or detecting the cause [2]. Accumulation of Hcy in organisms can occur as a consequence of endogenous factors, such as abnormalities of genes encoding the synthesis of enzymes involved in Hcy metabolism, but also exogenous factors such as deficiency of vitamins B6, vitamin B12, folate and/or diet rich in methionine [3]. Depending on the level of Hcy in serum, hyperhomocysteinemia (HHcy) can be defined as mild (15–30 μmol/L), moderate (30–100 μmol/L) and severe (>100 μmol/L) [4].

Data from the literature indicate an association between HHcy and endothelial dysfunction [5,6]. Studies on animal models show that elevated Hcy levels lead to accelerated development of endothelial dysfunction in mice [7,8]. This finding is also confirmed by studies in humans. Certain studies reported a negative effect of HHcy on endothelium-dependent vasodilation [9] but also its improvement after reduction of Hcy concentration [10,11]. Moreover, Hcy does not only have an effect on endothelial cells, but can also have an impact on macrophage lipid metabolism. Hcy promotes lipid accumulation and inhibits cholesterol efflux in macrophages by inhibiting liver X receptor alpha and decreasing expression of ATP binding cassette transporters A1 and G1, which are responsible for cholesterol efflux [12]. Accumulation of cholesterol in macrophages induces their transformation into foam cells and promotes development of atherosclerotic plaques [13]. Several studies have shown a connection between elevated plasma Hcy levels and changes in cardiac structure and function. The effects of homocysteine on the heart muscle are considered to be indirect and caused by induction of atherothrombotic lesions in coronary circulation [14], followed by the reduction of myocardial perfusion. Likewise, atherosclerotic changes in peripheral arterial blood vessels caused by endothelial dysfunction with impaired vasodilatory response (and following vasoconstrictory predomination) and vascular remodeling, can lead to the development of systemic hypertension [15,16,17,18]. Hcy levels show positive correlation with vasoconstrictors like endotelin-1 and thromboxane [19]. If this persists for a long time, increased arterial blood pressure increases afterload, and can cause remodeling of myocardium leading to its insufficiency. Effects of Hcy on cardiac tissue are mainly caused by remodeling of coronary arteries and fibrosis in cardiac tissue, especially in the form of perivascular interstitial fibrosis [20,21]. Patients with elevated Hcy have greater risk for all-cause mortality and major cardiovascular and cerebrovascular adverse events after percutaneous coronary intervention [22]. A recent meta-analysis showed that high Hcy levels are associated with higher risk for development of atrial fibrillation, which indicates that Hcy can also affect heart rhythm [23].

Another way in which Hcy can achieve its effects on the cardiac tissue is based on the induction of matrix metalloproteinases (MMP). Extracellular remodeling, which has been confirmed in CVD, largely depends on MMP-2 and MMP-9 activation. In an intact myocardium, MMPs are poorly active, while on the other hand, during HHcy their increased activity has been reported [24], although the exact mechanism of their activation has yet to be elucidated. Hcy metabolism also has an important role in epigenetic regulation of gene expression via DNA methylation because S-adenosyl-methionine, a precursor for Hcy synthesis in the re-methylation cycle, acts like a methyl group donor for organic molecules in biological systems, and epigenetic regulation of gene activity can be provided via methylation of thymine nucleotides in the DNA chain [25].

The results of various experimental studies have unequivocally proven that physical activity reduces the risk of developing cardiovascular and cerebrovascular diseases [26]. However, it remains unclear whether, and if, in what way aerobic physical activity affects Hcy level in blood [27]. Over the last few years, various studies have tried to elucidate the mechanism of this impact, however, the results have repeatedly been contradictory or inconsistent. Since the applied methodology differed significantly in these studies, a definite consensus was not reached. Some studies have reported a decrease in the level of Hcy in serum after aerobic physical activity [28,29], others have shown a correlation between intense physical activity and cardiorespiratory fitness on the one hand, and low levels of serum homocysteine on the other [30,31]. However, transient HHcy has been reported in several studies as a direct consequence of acute aerobic physical activity [32,33,34].

Taking all of these findings into consideration, the aim of the study was to investigate the effects of homocysteine load and four weeks of aerobic physical activity on cardiac tissue matrix metalloproteinase, lactate dehydrogenase and malate dehydrogenase enzymes activities, as well as on hepato-renal biomarkers in sera of experimental animals.

## 2. Results

### 2.1. Body Mass of Experimental Animals

During the entire experimental protocol, there were no lethal cases. At the beginning of the experimental protocol, animals from the active groups (CPA and HPA) had lower body masses in comparison to animals in the C and H group (Figure 1). After two weeks there were no differences in body mass between active and sedentary animals, but the body mass of animals in the CPA group were significantly higher in comparison to the HPA group and this difference remained present until the end of the experimental period (*p* < 0.05).

### 2.2. Biochemical Parameters in Sera of Rats

Biochemical parameters in sera of rats are presented in Table 1. Hcy level in serum of experimental animals was significantly higher in the HPA group (18.5 ± 5.1 μmol/L) in comparison to CPA group (12.9 ± 2.3 μmol/L), *p* < 0.05. Concentration of Hcy in serum was higher in the H (19.6 ± 8.2 μmol/L) and HPA group than in the C group (13.6 ± 3.8 μmol/L), but without reaching statistical significance. Glucose level in serum was significantly higher in both active groups (CPA: 5.7 ± 0.6 mmol/L; HPA: 5.8 ± 0.6 mmol/L) in comparison to sedentary groups (C: 2.4 ± 1.3 mmol/L, *p* < 0.01 vs. CPA and *p* < 0.01 vs. HPA; H: 2.9 ± 0.8 mmol/L, *p* < 0.01 vs. CPA and *p* < 0.01 vs. HPA), while the level of glucose in serum did not differ in the HPA group compared to CPA.

Total protein concentration was higher in the CPA group (62.0 (62.0–62.7) g/L) compared to the C (29.5 (24.0–33.0) g/L, *p* < 0.01) and H group (42.0 (26.0–45.0) g/L *p* < 0.01); also, concentration of proteins was higher in the HPA group (61.5 (59.2–62.0) g/L) than in both C (*p* < 0.01) and H (*p* < 0.01) groups. Concentration of serum albumin was increased in the CPA group (32.5 (31.2–33.0) g/L) in comparison to the C group (15.0 (12.3–15.7) g/L, *p* < 0.01), and the H group (21.0 (13.0–22.0) g/L, *p* < 0.01), similarly higher level of albumin was also determined in the HPA group (31.0 (31.0–32.7) g/L) compared to passive groups (C, *p* < 0.01; H, *p* < 0.01).

Determination of aminotransferases (ALT and AST) activities in rat sera showed the following results: the activity of ALT in serum was significantly increased after application of Hcy s.c. for two weeks (H; 123 (116–125) U/L) in comparison to the control group (C; 105 (91–110) U/L, *p* < 0.05). ALT activity was higher in the CPA group (213 (195–225) U/L) than in the C group (*p* < 0.01) and H group (*p* < 0.01). Determination of ALT activity in the HPA group has also shown increased values in comparison to both C (*p* < 0.01) and H (*p* < 0.01) groups. AST activity was increased in both active groups (CPA: 52 (47–58) U/L; HPA 57 (53–59) U/L) in comparison to group C (26 (20–34) U/L, *p* < 0.01 vs. CPA and *p* < 0.01 vs. HPA) and H group (35 (30–36) U/L, *p* < 0.01 vs. CPA and *p* < 0.01 vs. HPA).

Concentrations of urea and creatinine in serum were determined as parameters of renal function. The concentration of urea was higher in the CPA group (8.4 ± 1.6 mmol/L) in comparison to the C group (5.8 ± 1.8 mmol/L, *p* < 0.01) and H group (5.9 ± 1.4 mmol/L). Level of urea was higher in the HPA group (8.2 ± 1.1 mmol/L) than in the C (*p* < 0.01) and H group (*p* < 0.01). Creatinine level in serum was significantly increased in the CPA group (42.0 ± 2.4 μmol/L) compared to the C group (22.5 ± 4.8 μmol/L, *p* < 0.01) and H group (27.1 ± 6.3 μmol/L, *p* < 0.01). The similar results’ significance was detected in the HPA group, in which concentration of creatinine in serum (42.0 ± 1.6 μmol/L) was higher in comparison to both the C (*p* < 0.01) and H (*p* < 0.01) group.

Activity of amylase in serum was significantly higher in both active groups (CPA: 1661 (1517–1932) U/L; HPA: 1553 (1341–1802) U/L) in comparison to sedentary groups (C: 845 (814–893) U/L, *p* < 0.01 vs. CPA and *p* < 0.01 vs. HPA; H: 1063 (611–1143) U/L, *p* < 0.01 vs. CPA and *p* < 0.01 vs. HPA).

### 2.3. Total LDH and MDH Activities in Cardiac Tissue

The total activities of lactate dehydrogenase and malate dehydrogenase in cardiac tissue of experimental animals are presented in Figure 2.

Total activity of LDH was significantly increased in the HPA group (10.040 ± 1.154 mU/mg protein) in comparison to all groups (C: 6.402 ± 0.759 mU/mg protein, *p* < 0.01; H: 6.142 ± 0.321 mU/mg protein, *p* < 0.01; CPA: 5.708 ± 0.468 mU/mg protein, *p* < 0.01) (Figure 2A).

Total activity of MDH was higher in the HPA group compared to other groups, but without reaching statistical significance (Figure 2B).

### 2.4. Relative Activities of LDH and MDH Isoform in Cardiac Tissue

In direct electrophoretic zymography, four isoforms of LDH were detected in rat heart tissue homogenates in all groups (LDH1, LDH2, LDH3 and LDH4) (Figure 3). Densitometric analysis showed that activity of LDH 1 isoform was significantly increased in the HPA group (535.8 (436.3–650.1) px/mg protein) in comparison to the C and H group (151.2 (118.8–162.6) px/mg protein, *p* < 0.01; 153.1 (139.1–175.8) px/mg protein, *p* < 0.01, respectively) (Figure 4A). LDH2 activity was significantly higher in active groups, CPA (479.2 ± 100.5 px/mg protein) and HPA (774.0 ± 131.2 px/mg protein) in comparison to passive groups C and H (279.6 ± 38.2 px/mg protein, *p* < 0.05 vs. CPA, *p* < 0.01 vs. HPA; 274.7 ± 26.1 px/mg protein, *p* < 0.05 vs. CPA, *p* < 0.01 vs. HPA, respectively). When active groups were compared to each other, the activity of LDH2 isoform was significantly higher in the HPA group, *p* < 0.05 (Figure 4B). No significant difference between groups was detected in the activity of LDH3 isoform (Figure 4C). LDH4 activity was increased in the CPA group (262.5 (235.1–266.7) px/mg protein) compared to the C (78.9 (55.3–105.2) px/mg protein, *p* < 0.05) and H group (80.6 (52.3–103.3) px/mg protein, *p* < 0.05), likewise, LDH4 activity was significantly higher in the HPA group (369.7 (210.9–489.3) px/mg protein) than in the C (*p* < 0.01) and H group (*p* < 0.01) (Figure 4D).

In direct electrophoretic zymography, three isoforms of MDH were detected in rat heart tissue homogenates in all groups: cytosolic malate dehydrogenase (cMDH), mitochondrial malate dehydrogenase (mMDH) and peroxisomal malate dehydrogenase (pMDH) (Figure 5). Activity of cMDH was reduced in the CPA group (40.4 (32.2–44.4) px/mg protein) in comparison to the C (68.5 (62.2–83.7) px/mg protein, *p* < 0.05) and H group (74.8 (66.8–75.6) px/mg protein, *p* < 0.05) (Figure 6A). Similarly, densitometric analysis showed that mMDH activity was significantly decreased in the CPA group (207.4 ± 26.6 px/mg protein) in comparison to the C (326.5 ± 24.6 px/mg protein, *p* < 0.01) and HPA (365.1 ± 42.2 px/mg protein *p* < 0.01) group (Figure 6B). Values of pMDH activity did not differ between groups (Figure 6C).

### 2.5. Relative Activities of MMP-2 and MMP-9 in Cardiac Tissue

In SDS–PAGE zymography, in rat heart tissue homogenates in the C and H group only MMP-9 isoform was detected, while in the CPA and HPA groups, both MMP-2 and MMP-9 isoforms manifested their activities on gel slab (Figure 7A). There was no significant difference in MMP-2 activity between the CPA and HPA group (Figure 7B). MMP-9 activity was very low in passive groups C and H in comparison to active groups CPA and HPA. MMP-9 activity in the CPA group (40.2 (28.7–260.7) px/mg protein) was significantly elevated compared to the C group (6.3 (3.5–10.8) px/mg protein, *p* < 0.01) and H group (5.4 (3.7–8.9) px/mg protein, *p* < 0.01), likewise, MMP-9 activity in the HPA group (62.2 (23.5–228.9) px/mg protein) in comparison to the C (*p* < 0.01) and H group (*p* < 0.01) (Figure 7C).

## 3. Discussion

According to the World Health Organization, CVDs are the number one cause of death globally and it is expected that by 2030, almost 23.6 million people will die from CVDs, mainly from heart disease and stroke; however, HHcy occurs in about 5–7% of the world population [35,36]. As previously mentioned, it is one of the risk factors or markers for CVD, so taking that into consideration, explaining the connection between high Hcy levels in blood and CVD progression is an important research task.

This study was primarily focused on the effects of Hcy load on heart tissue as well as the effects of physical activity on these conditions. The main parameters measured in the cardiac tissue of experimental animals were the activities of enzymes included in energy metabolism: malate dehydrogenase (an enzyme in the Krebs cycle—aerobic metabolism) and lactate dehydrogenase, which is included in anaerobic metabolism, as well as the activities of enzymes involved in the remodeling of tissue—two isoforms of matrix metalloproteinases (MMP-2 and MMP-9). Total activities of LDH and MDH, and also the profile of their isoforms were determined in heart tissue homogenates. According to our findings, total activity of LDH was increased in the HPA group in comparison to all other groups. The determination of the activities of different LDH isoforms showed that in HPA group isoforms LDH1, LDH2 and LDH4 were significantly more active compared to the C and H group while the activity of isoform LDH3 did not differ significantly between groups. LDH2 was also more active in the HPA group compared to the CPA group. Electrophoretic zymography showed that mMDH isoform activity was increased in the HPA group compared to CPA. Besides this finding, isoforms mMDH and cMDH in the CPA group showed decreased activity in comparison to the C group, while the activity of pMDH did not differ significantly between groups.

Data from the literature suggest that Hcy load may affect the activity of enzymes involved in energy metabolism. An experimental study on rats by Kolling et al. investigated the effect of chronic HHcy on Krebs cycle enzymes’ activities [37]. According to the results of this study, Hcy did not independently induce a change in the activity of the enzymes involved in the Krebs cycle: citrate synthase, isocitrate dehydrogenase, and malate dehydrogenase. However, in combination with creatine application, Hcy led to a change in the activity of citrate synthase and isocitrate dehydrogenase. The synthesis of creatinine and Hcy are most likely metabolically related, thus, Wyss et al. [38] have shown that endogenous creatinine synthesis is the most important factor in maintaining the balance between methylation and Hcy formation. To date, adjuvant creatine therapy may correct the symptoms of energy imbalance due to HHcy. It is possible that in accordance with these results and in our study, Hcy did not independently lead to a change in the activity of enzymes involved in energy metabolism in comparison to the control group, however, under conditions of physical activity, Hcy can affect the change in activity of these enzymes.

The key metabolic reactions involved in energy production, such as the Krebs cycle, oxidative phosphorylation and electron transport chain (ETC) take place in mitochondria. Many studies have confirmed that HHcy can impact mitochondrial respiration by changing the activity of enzymes involved in these reactions [39]. Both in vitro and in vivo studies have showed that Hcy can reduce the activities of ETC complexes II, III and IV and decrease energy production in cardiac tissue [40,41,42]. Similar findings were also described in brain cortex and subcortical structures (amygdala, substantia nigra) [43,44,45]. The activity of succinate dehydrogenase, an enzyme in the Krebs cycle that is bound to the inner membrane of mitochondria, was decreased in cardiac tissue after acute application of Hcy on heart slices for 1 h [40]. On the other hand, chronic treatment with Hcy led to increased activity of succinate dehydrogenase in rat hearts [42]. In our study, application of Hcy for two weeks under conditions of physical activity increased activity of mitochondrial isoform of malate dehydrogenase. Timkova et al. described adaptive upregulation of isocitrate dehydrogenase in the cardiac tissue of rats with mild HHcy [41].

Taking into consideration all of these effects of HHcy on components of the electron transport chain in inner membrane of mitochondria, changes in the activities of enzymes involved in energy metabolism could be explained as a compensatory mechanism that aims to maintain the production of ATP at an adequate level, which could be the possible explanation for increasing mMDH activity in the HPA group in this study. It is interesting that the total activity of LDH was also increased in the HPA group although the model of physical activity used in this study is the model that is proven to activate aerobic metabolism, and conversion of pyruvate to lactate catalyzed with LDH is a reaction of anaerobic metabolism. We can assume that the increased total activity of LDH in the HFA group could be a compensatory response that directs energy metabolism towards anaerobic reactions because of the previously mentioned effects of HHcy on the respiratory chain in mitochondria [46,47,48,49]. The exact mechanism of these metabolic effects of Hcy should be investigated in new studies that focus on measuring the activity of other enzymes of aerobic metabolism in such conditions.

If we keep in mind that LDH is known as a biochemical marker of tissue damage, its increase could be explained, at least in part, by Hcy-mediated oxidative stress. Furthermore, high oxygen uptake during a four-week period of aerobic training is potentially associated with enhanced production of reactive oxygen species, contributing to heart tissue damage and LDH increase. Similar to our results, increased levels of LDH were reported during ischemia reperfusion injury, even though these effects were attenuated by high-intensity interval training on the 1st and 7th day post exercise [50]. Although there is a widespread belief that physical activity has beneficial effects on heart tissue, it seems that both duration and intensity of exercise are significant factors in accomplishing exercise-induced cardio protection [51].

Data from the literature indicate that Hcy can induce the formation of free radicals, and lead to the activation of MMPs. It is reported that Hcy stimulates accumulation of mast cells and interstitial collagen, leading to interstitial and perivascular fibrosis and disruption of the collagen/elastin ratio, which is of great importance for the contractile ability of the myocardium [52]. These changes, as the ultimate outcome, weaken the contractile efficiency of heart muscle. The activity of MMPs can cause damage to the gap junctions and cause interference in the conduction of impulses between cardiomyocytes, consequently leading to the development of arrhythmias [53]. It is reported that in an intact myocardium, MMPs are poorly active, while during HHcy their activity increases [24]. In our study, the independent application of Hcy did not increase MMP activity and no significant differences in the activities of MMP-9 were found in the H group in comparison to control. On the other hand, in both sedentary groups C and H, isoform MMP-2 did not express its activity. Our results showed that only physical activity was a significant factor for MMP activation since in both active groups MMP-2 expressed its activity in the gel slab. The activity of MMP-2 in the HPA group was slightly higher than in the CPA group, but this difference did not show statistical significance. MMP-9 activity was very low in passive groups (C and H) in comparison to active groups (CPA and HPA), but there was no difference in its activity between the H and C group, or between the CPA and HPA group.

Analyses of biochemical parameters in serum showed that concentrations of glucose, total proteins and albumin were significantly higher in active than in sedentary groups. This may be due to increased intake of food and the increased metabolic needs of active animals.

The activities of aminotransferases, ALT and AST were measured in order to estimate liver function. Results of this study showed increased AST activity in serum of animals in the H group in comparison to control. Likewise, the activities of both AST and ALT were increased in active groups compared to passive groups. Many studies have demonstrated that increased methionine blood concentration can impact function and cause injuries in peripheral tissues [54]. The possible explanation for these findings lies in the conversion of methionine to Hcy in the re-methylation cycle and its effect on increasing oxidative stress. Yamada et al. [55] reported that methionine load in the diet alters lipid metabolism in hepatocytes, causes oxidative stress, and induces hepatocytes damage. Treatment of rats with methionine can also lead to morphological changes in liver tissue such as rare focal necrosis of hepatocytes and periportal mononuclear infiltration [56].

Data from the literature recognize that HHcy can also affect renal function. It is reported in several studies that levels of Hcy in blood directly correlate with levels of creatinine in serum and glomerular filtration rate [57,58,59]. Chronic uremia also impairs the extrarenal activity of the enzymes responsible for Hcy metabolism [60]. In our study, higher concentrations of urea and creatinine were detected in the CPA and HPA groups in comparison to the C and H group.

## 4. Materials and Methods

### 4.1. Animal Ethics Statement

This study was approved by the Ethical Council for the Welfare of Experimental Animals, Ministry of Agriculture, Forestry and Water Management, Veterinary Directorate, Republic of Serbia (No: 323-07-02523/2018-05). All experimental procedures were done in accordance with the prescribed legislation (EU Directive for the Protection of Vertebrate Animals Used for Experimental and Other Scientific Purposes 86/609/EES) and ethical principles.

### 4.2. Experimental Animals

In this research, experiments were provided on male *Wistar albino* rats with a body mass of approximately 180 g at the beginning of the experimental protocol. Experimental animals were acclimatized and housed in standard laboratory conditions at an ambient temperature 20 ± 2 °C, 50 ± 15% air humidity and in a 12/12-h light/dark cycle, with the light period beginning at 07:30. The rats were housed in transparent plexiglass cages with a woodchip floor and were provided with a commercial rat pellet diet and water ad libitum for 3 days of adaptation period and throughout the four-week study period.

### 4.3. Experimental Protocol

This study was conducted over 4 weeks. Animals were acclimatized to laboratory conditions for 3 days. They were distributed into four groups with 10 animals per group: the control group (C) was treated with saline (0.9% NaCl 0.2 mL/day s.c.); homocysteine group (H) was treated with homocysteine (0.45 µmol/g b.w./day s.c.) [61]; physically active control group (CPA) was treated with saline (0.9% NaCl 0.2 mL/day s.c.) and a program of physical activity on a treadmill; and the physically active homocysteine group (HPA) was treated with homocysteine (0.45 µmol/g b.w./day s.c.) and a program of physical activity on a treadmill. Subcutaneous injection of substances was applied 2 times a day at intervals of 8 h during the first two weeks of experimental protocol.

Standard treadmill apparatus (Lab Animal-Treadmill, Elunit, Serbia) for small laboratory animals with four separated tracks was used in the study. Animals ran in the opposite direction to the treadmill belt movement, escaping the stimulating electrodes at the end of the tracks. Animals in the physically active groups, CPA and HPA, were prepared for the treadmill program: they practiced how to run on a treadmill for 3 consecutive days, for 10 min each day with the treadmill belt speed set at 10 m/min and 0° incline. After adaptation, within the experimental protocol, animals in the CPA and HPA groups were made to run on the treadmill apparatus for 30 min/day for 28 consecutive days. Belt speed was set at 20 m/min with 0° incline. One experimental study on rats [62] recognized that the highest exercise intensity at which blood lactate does not increase beyond the initial transient during a constant exercise load was obtained at a belt speed of 20 m/min for treadmill running rats, indicating that this protocol can be considered as aerobic physical activity.

After 14 days of experimental protocol, animals from the sedentary groups (C and H) were sacrificed, and after 28 day of experimental protocol animals from active groups (CPA and HPA) were sacrificed by decapitation. Blood for biochemical analyses of hepatorenal biomarkers in serum was collected in appropriate vacutainers through a glass funnel and after 15 min at room temperature it was centrifuged (15 min, 3000 rpm). The hearts of animals were homogenized on ice in buffer (20 mmol/L Tris-HCl, pH 7.5, 250 mmol/L sucrose, 1% Triton X-100, 1 mmol/L protease inhibitor phenylmethylsulfonyl fluoride (PMSF) and 1 μg/mL leupeptin) [63] in a ratio of 1:10, homogenates were centrifuged (10 min, 10,000 rpm). The total protein concentration in the obtained supernatants was measured by the Bradford method [64]. Supernatants were kept at −20 °C for biochemical analyses of enzymes activities in cardiac tissue.

### 4.4. Biochemical Analyses in Serum

The serum glucose level, total proteins, albumin, liver enzymes activity: aspartate aminotransferase (AST) and alanine aminotransferase (ALT), renal parameters (urea and creatinine) and alpha-amylase (AMY) were analyzed using spectrophotometry and commercial kits (Siemens Healthcare Diagnostics Inc., Newark, NJ, USA) on an automatic biochemical analyzer (Dimension Xpand, Siemens Healthcare Diagnostics Inc., Newark, NJ, USA).

### 4.5. Determination of Total LDH and MDH Activities in Cardiac Tissue

Activity of LDH was determined by spectrophotometric measurement of the absorbance decrease at 340 nm during oxidation of NADH in reaction: pyruvate + NADH + H^+^ ⇔ lactate + NAD^+^ [65]. The reaction was started by adding substrate Na-pyruvate 0.1 mL, 23 mmol/L to reaction mixture: 2.9 mL phosphate buffer 0.1 mM, pH = 7.0; NADH 0.05 mL, 14 mmol/L (prepared just before analysis and kept on ice during usage because of NADH instability) and sample (supernatant of cardiac tissue homogenate) 0.01 mL. One unit of LDH activity (U) catalyzes the transformation of 1 μmol of NADH per minute under the test conditions.

Determination of MDH activity was based on measuring the absorbance decrease during oxidation of NADH in the reaction: oxalacetate + NADH + H^+^ ⇔ malate + NAD^+^ at 340 nm [66]. The reaction was started by adding substrate Na_2_-oxaloactetate 0.1 mL, 15 mmol/L to reaction mixture: 2.9 mL phosphate buffer 0.1 mM, pH = 7.5; NADH 0.05 mL, 14 mmol/L and sample (supernatant of cardiac tissue homogenate) 0.01 mL. One unit of MDH activity (U) catalyzes the transformation of 1 μmol of NADH per minute under the test conditions.

For the determination of total activities of LDH and MDH, absorbance decrease was measured on a UV-2600 spectrophotometer (Shimadzu, Japan).

### 4.6. Determination of Relative Activities of LDH and MDH Isoform in Cardiac Tissue by Electrophoresis and Densitometric Analysis

Different isoforms of LDH in cardiac tissue were detected by the direct electrophoretic zymography. Electrophoresis was performed on 7.5% native polyacrylamide gel. Isoforms was visualized during incubation of gel in reaction mixture (Li-lactate 0.25 g, NAD 18.75 mg, phenazinmethosulphate (PMS) 0.625 mg, nitrobluetetrazolium (NBT) 12.5 mg and TRIS buffer 25 mL, pH 8.3) for 10 min at 37 °C in darkness. Isoforms appeared on the gel as dark blue bands of formazan, which is formed after the reduction of NBT in the presence of the electron transfer mediator PMS and NAD as a coenzyme [67].

Different isoforms of MDH in cardiac tissue were determined by the direct electrophoretic zymography. Electrophoresis was performed on 7.5% native polyacrylamide gel. Isoforms was visualized during incubation of gel in reaction mixture (Na_2_-malate 0.25 g, NAD 15 mg, phenazinmethosulphate (PMS) 0.5 mg, nitrobluetetrazolium (NBT) 10 mg and phosphate buffer 25 mL pH = 7.1) for 10 min at 37 °C in darkness. After visualization, isoforms of MDH appeared as dark blue bands of formazan, which is formed as a product of the reduction of nitroblue tetrazolium in the presence of phenazinemetosulphate as an electron transfer mediator and NAD as a coenzyme [68].

LDH and MDH isoforms on the obtained zymograms were analyzed using the ImageJ Q.42 software package. Their relative activities were determined by measuring the surface under the densitometric curves in pixels, and values were estimated for the protein concentration in the sample (px/mg protein).

### 4.7. Determination of Relative Activities of MMP-2 and MMP-9 in Cardiac Tissue by Electrophoresis and Densitometric Analysis

Relative activities of MMP isoforms were evaluated by sodium dodecyl sulfate–polyacrylamide gel electrophoresis (SDS–PAGE) [69]. Cardiac tissue samples were diluted 20 times in 200 g/L sucrose to prepare the test samples, which were incubated for 40 min at 37 °C. For each sample, 6.25 µg of total tissue protein was loaded. Electrophoresis was performed on 7.5% polyacrylamide gels copolymerized with 0.1% gelatin (substrate for MMP) at 4 °C at a constant current of 50 mA. When the tracking dye at the front reached the bottom of the gel, the gel was removed and shaken gently for 45 min in 0.25 g/L Triton X-100 to remove the SDS, then the gel slabs were transferred to a bath with deionized water and washed twice for 20 min to remove the Triton X-100. After that the gels were incubated and shaken for 24 h in 0.1 mol/L glycine, 50 mmol/L Tris–HCl, 5 mmol/L CaCl_2_, 1 mol/L ZnCl_2_, and 0.5 mol/L NaCl, at pH 8.3 and 37 °C. Gels were then stained for 3 h with Coomassie brilliant blue G-250. After visualization, gels were scanned and densitometric analysis was performed to measure the relative activity of the MMP isoforms using ImageJ Q.42 software package. The degree of gelatin digestion was quantified as enzyme activity by measuring the surface under the densitometric curves in pixels. Densitometric data were then normalized for protein concentration in sample (px/mg protein).

### 4.8. Chemicals

All chemicals were obtained from Sigma-Aldrich Chemie GmbH, Schnelldorf, Germany.

### 4.9. Statistical Analysis

SPSS 19.0 for Windows software package was used for statistical analyses. Distribution of data was tested using the Shapiro–Wilk test. Parameters with normal distribution were expressed as mean ± SD, while parameters with non-normal distribution were expressed as median with interquartile range (25th percentile–75th percentile). Homogeneity of variances was tested using Levene’s test. Statistical comparisons between the experimental groups were conducted using ANOVA with Tukey’s post hoc test, Welch’s ANOVA with Games–Howell post hoc test or Kruskal–Wallis followed by Dunn–Bonferoni post hoc test and Mann–Whitney U test where appropriate. Differences were considered significant at *p* < 0.05.

## 5. Conclusions

The aim of this study was to investigate the possible effect of independent application of homocysteine as well as its effect under the condition of aerobic physical activity on the activities of enzymes, i.e., MMP-2 and MMP-9, LDH and MDH in cardiac tissue and on hepato-renal biochemical parameters in sera of rats. It was demonstrated that the increased of total activitiy of LDH (mainly caused by the higher activitiy of isoform LDH2) and activity of mMDH were found under the condition of homocysteine-treated rats plus aerobic physical activity. Independent application of Hcy did not lead to these changes. Physical activity leads to activation of MMP-2 isoform and to increased activity of the MMP-9 isoform in both Hcy-treated and control rats.

## Figures and Tables

**Figure 1 ijms-22-06792-f001:**
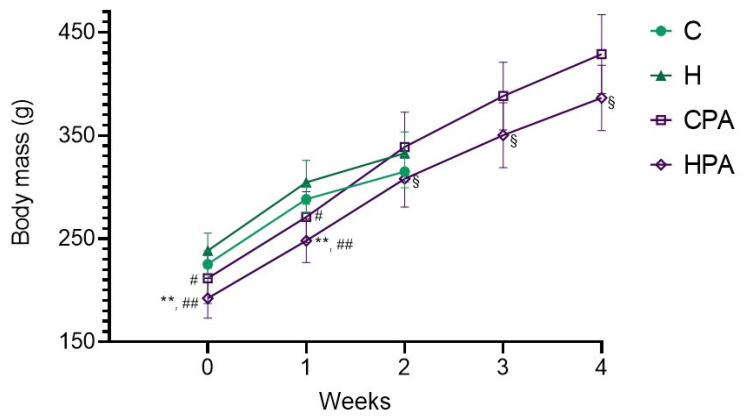
Body mass of experimental animals during four weeks of experimental period. C—0.9% NaCl; H—homocysteine; CPA—0.9% NaCl + physical activity on treadmill; HPA—homocysteine + physical activity on treadmill. ** *p* < 0.01 vs. C group; # *p* < 0.01 vs. H group, ## *p* < 0.01 vs. H group, § *p* < 0.05 vs. CPA group. One-way ANOVA with Tukey’s post hoc test and T-test for independent samples.

**Figure 2 ijms-22-06792-f002:**
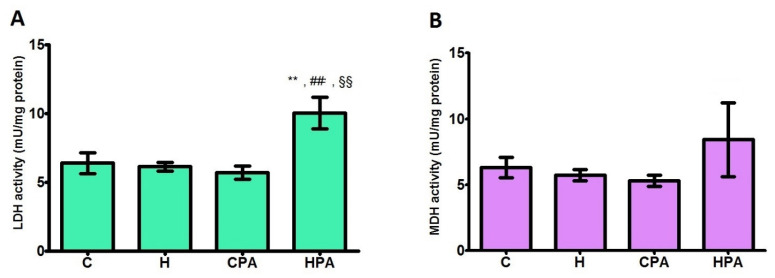
Total activities of lactate dehydrogenase (**A**) and malate dehydrogenase (**B**) in rat cardiac tissue. C—0.9% NaCl; H—homocysteine; CPA—0.9% NaCl + physical activity on treadmill; HPA—homocysteine + physical activity on treadmill. (**A**): ** *p* < 0.01 versus C, ## *p* < 0.01 versus H, §§ *p* < 0.01 versus CPA. One-way ANOVA with Tukey’s post hoc test. (**B**): Welch’s ANOVA.

**Figure 3 ijms-22-06792-f003:**
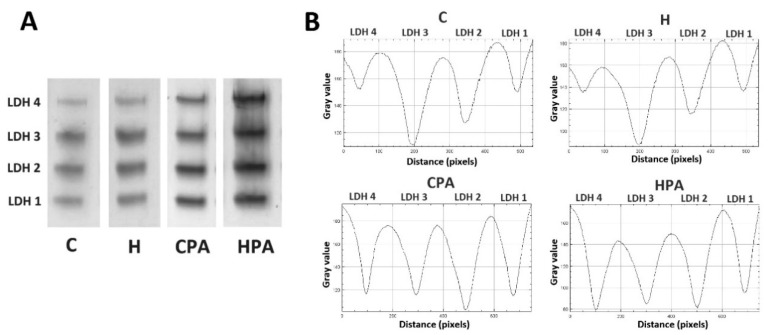
Representative zymograms lactate dehydrogenase isoforms (**A**) and representative densitometric curves lactate dehydrogenase isoforms (**B**) in rat cardiac tissue. C—0.9% NaCl; H—homocysteine; CPA—0.9% NaCl + physical activity on treadmill; HPA—homocysteine + physical activity on treadmill.

**Figure 4 ijms-22-06792-f004:**
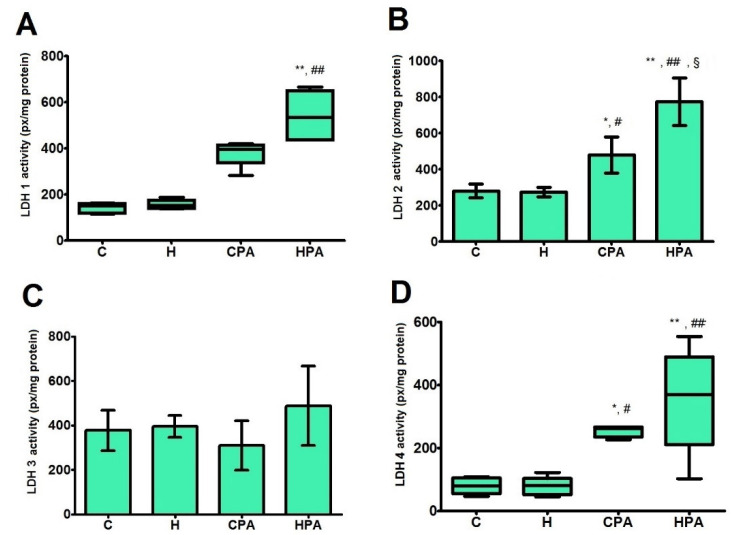
Relative activities of different isoforms of lactate dehydrogenase in rat cardiac tissue—densitometric analysis. C—0.9% NaCl; H—homocysteine; CPA—0.9% NaCl + physical activity on treadmill; HPA—homocysteine + physical activity on treadmill. (**A**,**D**): Kruskal–Wallis with Dunn–Bonferoni post hoc test—* *p* < 0.05 versus C, ** *p* < 0.01 versus C, # *p* < 0.05 versus H, ## *p* < 0.01 versus H. (**B**): Welch’s ANOVA with Games–Howell post hoc test—* *p* < 0.05 versus C, ** *p* < 0.01 versus C, # *p* < 0.05 versus H, ## *p* < 0.01 versus H, § *p* < 0.05 versus CPA. (**C**): One-way ANOVA.

**Figure 5 ijms-22-06792-f005:**
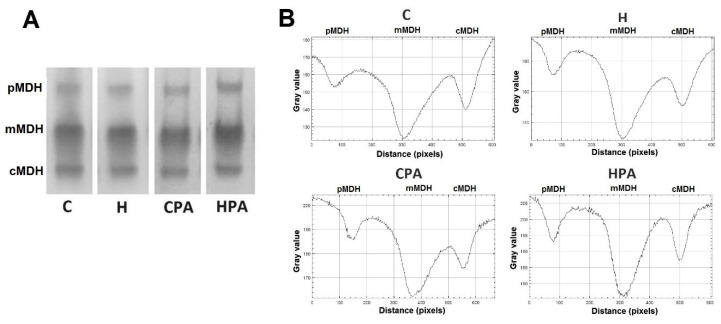
Representative zymograms of malate dehydrogenase isoforms (**A**) and representative densitometric curves malate dehydrogenase isoforms (**B**) in rat cardiac tissue. C—0.9% NaCl; H—homocysteine; CPA—0.9% NaCl + physical activity on treadmill; HPA—homocysteine + physical activity on treadmill. cMDH—cytosolic malate dehydrogenase, mMDH—mitochondrial malate dehydrogenase, pMDH—peroxisomal malate dehydrogenase.

**Figure 6 ijms-22-06792-f006:**
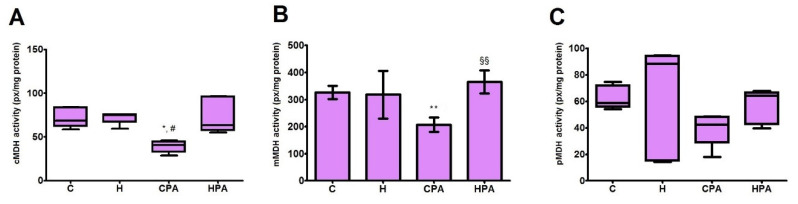
Relative activities of different isoforms of malate dehydrogenase in rat cardiac tissue—densitometric analysis. C—0.9% NaCl; H—homocysteine; CPA—0.9% NaCl + physical activity on treadmill; HPA—homocysteine + physical activity on treadmill. cMDH—cytosolic malate dehydrogenase, mMDH—mitochondrial malate dehydrogenase, pMDH—peroxisomal malate dehydrogenase. (**A**,**C**): Kruskal–Wallis with Dunn–Bonferoni post hoc test: * *p* < 0.05 versus C, # *p* < 0.05 versus H. (**B**): Welch’s ANOVA with Games–Howell post hoc test: ** *p* < 0.01 versus C: §§ *p* < 0.01 versus CPA.

**Figure 7 ijms-22-06792-f007:**
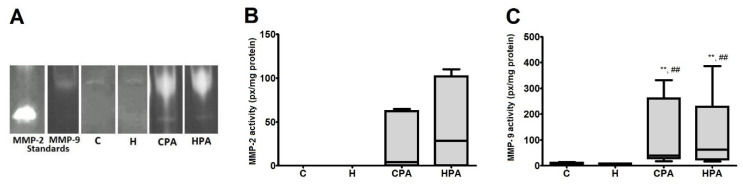
Representative zymograms of cardiac tissue matrix metalloproteinases (MMP-2 and MMP-9) activity detected by SDS–PAGE (**A**) and relative activities of MMP-2 (**B**) and MMP-9 (**C**) in rat cardiac tissue. C—0.9% NaCl; H—homocysteine; CPA—0.9% NaCl + physical activity on treadmill; HPA—homocysteine + physical activity on treadmill. Mann–Whitney U test: ** *p* < 0.01 versus C, ## *p* < 0.01 versus H.

**Table 1 ijms-22-06792-t001:** Biochemical parameters in sera of the experimental animals.

Parameters in Serum	C	H	CPA	HPA
Hcy (µmol/L) ^a^	13.6 ± 3.8	19.6 ± 8.2	12.9 ± 2.3	18.5 ± 5.1 §
Glucose (mmol/L) ^b^	2.4 ± 1.3	2.9 ± 0.8	5.7 ± 0.6 **, ##	5.8 ± 0.6 **, ##
Proteins (g/L) ^c^	29.5 (24.0–33.0)	42.0 (26.0–45.0)	62.0 (62.0–62.7) **, ##	61.5 (59.2–62.0) **, ##
Albumin (g/L) ^c^	15.0 (12.3–15.7)	21.0 (13.0–22.0)	32.5 (31.2–33.0) **, ##	31.0 (31.0–32.7) **, ##
AST (U/L) ^c^	105 (91–110)	123 (116–125) *	213 (195–225) **, ##	204 (196–235) **, ##
ALT (U/L) ^c^	26 (20–34)	35 (30–36)	52 (47–58) **, ##	57 (53–59) **, ##
Urea (mmol/L) ^b^	5.8 ± 1.8	5.9 ± 1.4	8.4 ± 1.6 **, ##	8.2 ± 1.1 **, ##
Creatinine (μmol/L) ^a^	22.5 ± 4.8	27.1 ± 6.3	42.0 ± 2.4 **, ##	42.0 ± 1.6 **, ##
Amylase (U/L) ^c^	845 (814–893)	1063 (611–1143)	1661 (1517–1932) **, ##	1553 (1341–1802) **, ##

C—0.9% NaCl; H—homocysteine; CPA—0.9% NaCl + physical activity on treadmill; HPA—homocysteine + physical activity on treadmill. Hcy—homocysteine; AST—aspartate aminotransferase; ALT—alanine aminotransferase. a—data are presented as mean ± standard deviation and were analyzed with Welch’s ANOVA followed by Games–Howell post hoc test. b—data are presented as mean ± standard deviation and were analyzed with one-way ANOVA followed by Tukey’s post hoc test. c—data are presented as median (25th percentile—75th percentile) and were analyzed with Kruskal–Wallis followed by Mann–Whitney U test. * *p* < 0.05 vs. C group; ** *p* < 0.01 vs. C group; ## *p* < 0.01 vs. H group; § *p* < 0.05 vs. CPA.

## Data Availability

The data used to support the findings of this study are available from the corresponding author upon request.

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
