# Peer review of "Four Weeks of Aerobic Training Affects Cardiac Tissue Matrix Metalloproteinase, Lactate Dehydrogenase and Malate Dehydrogenase Enzymes Activities, and Hepatorenal Biomarkers in Experimental Hyperhomocysteinemia in Rats"

_ijms, 2021, doi:10.3390/ijms22136792_

Round 1
Reviewer 1 Report
The authors of this manuscript investigated the influence of aerobic exercise on selected biochemical parameters in hyperhomocysteinic (HHcy) rats. This work is mainly descriptive showing increased activities of lactate dehydrogenase and malate dehydrogenase in cardiac tissue of exercised HHcy rats while activation of matrix metalloproteinase (MMP) isoforms MMP-2 and MMP-9 were similar in both exercised HHcy and exercised control groups. Serum biochemical parameters in exercised HHcy rats were altered in comparison to sedentary HHcy and control groups, but there were no significant differences between exercised HHcy and exercised control groups. The results may contribute to understanding the relation between hyperhomocysteinemia and physical training. However, there are following issues which need explanation and appropriate corrections.
Major concerns:
Discussion is not written well. It is mainly recapitulation of results with weak interpretation/comparison with published data. Parts of Discussion (e.g. lines 285-293, 318-333) are irrelevant to presented findings. Clear explanation what these findings actually mean is lacking.
Line 412: Design of study: Why animals from sedentary groups were sacrificed after 14 days and those from exercised groups after 28 days.
Author Response
Point 1: Discussion is not written well. It is mainly recapitulation of results with weak interpretation/comparison with published data. Parts of Discussion (e.g. lines 285-293, 318-333) are irrelevant to presented findings. Clear explanation what these findings actually mean is lacking.
Response 1: According to suggestions of reviewer 1 we changed discussion section by modifying paragraph (lines 285-293 in first version of manuscript). We added new references that more precisely describe effects of homocysteine on oxidative metabolism in mitochondria. We removed paragraph (lines 318-333 in first version of manuscript) because reviewer 1 considered it is not closely related to the topic of our research and our findings.
Point 2: Line 412: Design of study: Why animals from sedentary groups were sacrificed after 14 days and those from exercised groups after 28 days.
Response 2: Animals form sedentary groups were sacrificed after 14th day of experimental protocol to prove experimental model of hyperhomocysteinemia according to reference 62.
Reviewer 2 Report
In this paper the authors examine the effect of an application of homocysteine, and its interaction with physical activity, on the activity of MMP, LDH and MDH in rats. As far as concern the statistical analyses, my only concern refers to the use of ANOVA and Tukey post-hoc test. Both these approaches require the assumption of homogeneity of variance that, in some cases, does not appear to be confirmed in the data. Did the authors check this assumption before applying such procedure? Moreover, the authors use the Dunn's test as post-hoc after the Kruskal-Wallis omnibus test. Because this procedure did not account for multiplicity (as Tukey HSD does), did the author adopt a specific adjustment approach (i.e Holm, FDR etc.)?
Author Response
Point 1: As far as concern the statistical analyses, my only concern refers to the use of ANOVA and Tukey post-hoc test. Both these approaches require the assumption of homogeneity of variance that, in some cases, does not appear to be confirmed in the data. Did the authors check this assumption before applying such procedure?
Response 1: According to suggestions of reviewer 2 we performed Levene’s test to check homogeneity of variances of different parameters in our study and we compared means between groups using Welch’s ANOVA and Games-Howell post hoc test where appropriate.
Point 2: Moreover, the authors use the Dunn's test as post-hoc after the Kruskal-Wallis omnibus test. Because this procedure did not account for multiplicity (as Tukey HSD does), did the author adopt a specific adjustment approach (i.e Holm, FDR etc.)?
Response 2: In nonparametric analyses after significant Kruskal-Wallis test we used Dunn with Bonferoni correction post hoc test which takes in consideration number of comparisons made. We noted all of these changes in “Materials and methods – statistical analyses” section as well as in descriptions of figures and table.
Round 2
Reviewer 1 Report
The authors have answered the questions and made the revision according to suggestions. I recommend the manuscript for the publication.